# A Novel 5-Chloro-*N*-phenyl-1*H*-indole-2-carboxamide Derivative as Brain-Type Glycogen Phosphorylase Inhibitor: Potential Therapeutic Effect on Cerebral Ischemia

**DOI:** 10.3390/molecules27196333

**Published:** 2022-09-26

**Authors:** Yatao Huang, Shuai Li, Youde Wang, Zhiwei Yan, Yachun Guo, Liying Zhang

**Affiliations:** 1Laboratory of Traditional Chinese Medicine Research and Development of Hebei Province, Institute of Traditional Chinese Medicine, Chengde Medical University, Chengde 067000, China; 2Department of Pathogen Biology, Chengde Medical University, Chengde 067000, China

**Keywords:** brain-type glycogen phosphorylase inhibitor, mouse astrocytes, glycolysis, apoptosis, oxidative phosphorylation

## Abstract

Brain-type glycogen phosphorylase inhibitors are potential new drugs for treating ischemic brain injury. In our previous study, we reported compound **1** as a novel brain-type glycogen phosphorylase inhibitor with cardioprotective properties. We also found that compound **1** has high blood–brain barrier permeability through the ADMET prediction website. In this study, we deeply analyzed the protective effect of compound **1** on hypoxic-ischemic brain injury, finding that compound **1** could alleviate the hypoxia/reoxygenation (H/R) injury of astrocytes by improving cell viability and reducing LDH leakage rate, intracellular glucose content, and post-ischemic ROS level. At the same time, compound **1** could reduce the level of ATP in brain cells after ischemia, improve cellular energy metabolism, downregulate the degree of extracellular acidification, and improve metabolic acidosis. It could also increase the level of mitochondrial aerobic energy metabolism during brain cell reperfusion, reduce anaerobic glycolysis, and inhibit apoptosis and the expression of apoptosis-related proteins. The above results indicated that compound **1** is involved in the regulation of glucose metabolism, can control cell apoptosis, and has protective and potential therapeutic effects on cerebral ischemia-reperfusion injury, which provides a new reference and possibility for the development of novel drugs for the treatment of ischemic brain injury.

## 1. Introduction

Ischemic brain injury is the main clinical cause of death and disability; however, there is still no effective treatment [1]. Brain glycogen has a crucial role in hypoxic-ischemic brain injury [2]. Abnormal glycogen metabolism is an essential pathological factor in hypoxic-ischemic brain injury [3], and brain-type glycogen phosphorylase (PYGB) is a key enzyme that catalyzes brain glycogen metabolism [4]. Therefore, PYGB inhibitors are expected to become novel drugs for preventing and treating ischemic brain injury.

Brain glycogen is mainly found in astrocytes [5]. Neurons take up lactic acid released in astrocytes as their main energy metabolism substrate during neural activity, while intracellular glycogen, as an energy buffer system, allows astrocytes to display stronger glycolysis than the neurons’ solution [6,7,8]. Physiological stimulation is used to make the metabolic rate of brain glycogen reach 0.06–0.5 mmol/(g.min). This rate could be as high as 0.2–2.6 mmol/(g.min) when energy metabolism is disturbed and in the resting state (0.004–0.01 mmol/(g.min)), 6 to 50 times and more than 200 times, respectively [9]. It is speculated that the rapid decomposition of glycogen during the energy metabolism disorder is a major factor leading to the excessive production of lactic acid, tissue acidification, and metabolic acidosis in the early stage of ischemia. Therefore, improving glucose metabolism and controlling neuronal apoptosis is of great significance for improving cerebral ischemia injury [10]. PYGB is an important enzyme that catalyzes glycogen degradation and glycogen metabolism, whose pharmacological inhibition could be used to treat diseases related to abnormal glycogen metabolism.

Compound **1** is a novel GP inhibitor (Figure 1). In vitro enzyme activity analysis of compound **1** revealed that it has the best inhibitory activity against PYGB among the three isoforms of glycogen phosphorylase (IC_50_ of PYGB, PYGL, and PYGM, which are 90.27 nM, 1537.5 nM, and 144.21 nM, respectively). In a previous study, we conducted an in-depth analysis, which revealed that compound **1** has a cardioprotective effect [11]. At the same time, by using the ADMET prediction website, we found that compound **1** has high blood–brain barrier permeability (BBB: 3.9803). In this study, we further analyzed the effect of compound **1** on ischemic brain injury. According to a series of experiments, we preliminarily found the protective effect of compound **1** on cerebral astrocyte H/R injury and its possible mechanism, which provided novel insights into the therapeutic strategies for treating cerebral ischemia.

## 2. Results and Discussion

### 2.1. Protective Effect of Compound **1** on H/R Injury of Brain Astrocytes

In the present study, we mainly analyzed the effect of compound **1** on H/R injury of brain astrocytes. We choose nimodipine as positive control drug due to its therapeutic effect on cerebral ischemia, and it has been practically used in the clinical treatment field for many years [12,13,14]. After we isolated astrocytes successfully (Figure 2), the effect of compound **1** on astrocytes was tested using the CCK-8 method, which showed that the cell viability of the H/R group was significantly reduced, and the LDH leakage rate, ROS, and glucose levels were significantly increased compared with the blank group, thus indicating that the H/R model was successfully established (Figure 3). Moreover, compound **1** significantly improved cell viability compared with the H/R group (Figure 3A,B). Additionally, the LDH leakage rate was also remarkably reduced (Figure 3C).

Relevant studies have shown that ROS and glucose levels could effectively reflect ischemic brain injury [15,16,17,18]. The above results showed that compared with the H/R group, the intracellular glucose content (Figure 3D) and ROS levels were obviously decreased (Figure 3E,F), whereas the cell viability (1 μM), LDH leakage rate (3 μM), ROS (1 μM), and glucose content (1 μM) of the group with compound **1** reached a fair effect with the positive control drug nimodipine (NMP 10 μM). These results strongly implicated that compound **1** had a significant protective effect on cellular H/R injury.

### 2.2. Compound **1** Improves Mouse Astrocyte Energy Metabolism

Ameliorating glucose metabolism is essential for improving cerebral ischemia injury [19,20,21]. Previous research results showed that compound **1** interferes with glucose metabolism to a certain extent, reducing glucose content, while glucose metabolism was closely related to ATP level. In addition, additional inhibition of glucose metabolism would further reduce ATP levels [22,23,24]. Therefore, we further determined the effect of compound **1** on cellular ATP levels after H/R injury, finding that in contrast with the blank group, the H/R group significantly increased the ATP level. Meanwhile, compared with the H/R group, compound **1** dramatically reduced the level of cellular ATP, and the effect of compound **1** coincided with that of NMP (10 μM) at a concentration of 1 μM (Figure 4). As evidenced by these results, compound **1** remarkably improved energy metabolism in brain cells largely for the benefit of reducing anaerobic glucose metabolism.

### 2.3. Compound **1** Significantly Downregulates the Degree of Extracellular Acidification and Ameliorates Metabolic Acidosis

Previous studies have confirmed that anaerobic glycolysis is enhanced during cerebral ischemia, leading to the accumulation of lactate and pyruvate, which in turn causes acidosis [25]. Furthermore, lactic acid and its corresponding acidosis are the primary causes of astrocyte damage after cerebral ischemia [26]. Therefore, we further investigated whether the protective effect of compound **1** on brain astrocyte H/R injury is related to the inhibition of anaerobic glycolysis to produce excess lactate and ameliorate acidosis.

Seahorse Cell Energy Analyzer was used to measure the extracellular acidification rate (ECAR), namely, an indirect measure of lactate secretion and glycolysis [27]. As shown in Figure 5, compared with blank, the basal glycolysis level (Glycolysis), Glycolytic Capacity (Glycolytic Capacity), and Glycolytic Reserve (Glycolytic Reserve) of cells in the H/R group were all greatly reduced. The results showed that the level of energy metabolism provided by cell glycolysis was distinctly restrained. On the other hand, after administration of compound **1** and NMP, the basal glycolysis level (Glycolysis), Glycolytic Capacity (Glycolytic Capacity), and Glycolytic Reserve (Glycolytic Reserve) of cells were significantly increased relative to the H/R group. Even at the concentration of 1 μM, its effect was already comparable to that of NMP, which showed that compound **1** is better than NMP. Moreover, there was no significant difference in the level of non-glycolytic acidification, which indicated that compound **1** downregulates the degree of extracellular acidification and improves metabolic acidosis.

### 2.4. Compound **1** Significantly Increases Mitochondrial Aerobic Energy Metabolism and Decreases Anaerobic Glycolysis

In the presence of reperfusion, ROS is produced in large quantities by mitochondria, which aggravates cell damage and mitochondrial damage due to the respiratory burst [28,29]. Currently, the cellular oxygen consumption rate can evaluate mitochondrial function; thus, we measured the cellular oxygen consumption rate (OCR) adopted by Seahorse in order to explore the effect of compound **1** on cellular mitochondrial function and mitochondrial ROS production [30,31].

As shown in Figure 6, compared with the Blank group, the Basal Respiration (basal oxygen consumption capacity), Maximal Respiration (maximum oxygen consumption capacity), ATP Production (ATP synthesis amount), and Spare Respiratory (oxygen consumption potential) of the cell’s capacity in the H/R group were significantly lowered, indicating that the level of cellular mitochondrial oxidative phosphorylation aerobic respiration energy metabolism was significantly inhibited. For the H/R group, after compound **1** was combined with NMP, the basal oxygen consumption capacity (Basal Respiration), ATP production, maximum oxygen consumption capacity (Maximal Respiration), and oxygen consumption potential (Spare Respiratory Capacity) of cells were significantly improved. Additionally, when the concentration was 1 μM, the effect of compound **1** was comparable to that of NMP. Moreover, there was no significant difference between the levels of non-ATP synthesis oxygen consumption rate (Proton Leak) and non-mitochondrial oxygen consumption rate (Non-mitochondrial Respiration). The above results showed that compound **1** significantly promotes the levels of mitochondrial aerobic energy metabolism and reduces anaerobic glycolysis.

### 2.5. Proportion of Compound **1** Inhibits Apoptosis and the Expression of Apoptosis-Related Proteins

Apoptosis is considered the most critical link in cerebral ischemia injury and an important form of neuronal cell death during various cerebral ischemia processes [32,33]. We used Annexin VI to detect the apoptosis of the mitochondrial pathway after cell hypoxia reperfusion stress.

As shown in Figure 7A,B, the H/R and Vehicle groups significantly increased the proportion of upregulated cells compared with the blank group. Compared with the H/R group, compound **1** significantly inhibited the apoptosis rate, and its effect at a concentration of 3 μM was also consistent with that of NMP at 10 μM, which indicated that compound **1** is superior to NMP.

In addition, we further detected the expression of apoptosis-related proteins such as Caspase-3 and Bax in brain astrocytes by Western blot. As shown in Figure 7C, compound **1** significantly inhibited the expression of Cleaved Caspase 3 and Bax compared with the H/R group. Additionally, the inhibitory effect of compound **1** on the expression of Cleaved-Caspase 3 at a concentration of 3 μM was comparable to that of NMP at a concentration of 10 μM, and the inhibitory effect on the expression of Bax was more significant. The above results indicate that compound **1** can inhibit the proportion of apoptosis and the expression of apoptosis-related proteins and significantly affect the protection of brain cells.

## 3. Materials and Methods

### 3.1. Animals

Male C57BL/6J mice within 24 h of birth were obtained from Beijing Huafukang Biotecnology Co., Ltd. All animal studies (including the mice euthanasia procedure) were done in compliance with the regulations and guidelines of Chengde Medical University institutional animal care and conducted according to the Association for the Assessment and Accreditation of Laboratory Animal Care International (AAALAC) and the Institutional Animal Care and Use Committee (IACUC) guidelines.

### 3.2. Cell Culture

After the mice (C57) were sacrificed, they were immersed in 75% ethanol for disinfection. Their neck was clamped with vascular forceps from the back, and the scalp and skull were separated from the occipital region forward with ophthalmic scissors, after which they were peeled off with ophthalmic forceps and removed, including meninges and blood vessels. The brain (excluding the hippocampus) was placed in 1 × PBS (pH 7.4) and rinsed to remove the superficial blood capillaries until the brain tissue was milky white. Ophthalmic tweezers were used to shred the tissue into a powder shape. Ophthalmic scissors were then used to cut repeatedly for 10 min. Finally, a pipette gun was used to pipet 20 to 30 times gently. Next, we transferred the shredded tissue to a 15 mL centrifuge tube at 37 °C and digested it in a water bath for 15 min. A digestion stop solution was added at a ratio of 1:1 to stop the enzymatic reaction, after which cells were collected, resuspended, and centrifuged at 1000 rpm for 10 min. The supernatant was then discarded, and cells were mixed with added culture medium and counted.

The cell density was adjusted to inoculate the culture flask at 1 × 10^6^ cells/mL. Next, primary mouse brain astrocytes were cultured in DMEM medium, supplemented with 10% FBS, 1% penicillin/streptomycin in a humidified atmosphere containing 5% CO_2_/95% air at 37 °C.

### 3.3. Experiment Grouping

Cells were randomly divided into 5 groups: (1) Blank group; (2) Hypoxia/reoxygenation group (mice astrocytes were given hypoxia for 6 h and then were reoxygenated for 24 h); (3) Hypoxia/reoxygenation + solvent group; (4) Hypoxia/reoxygenation + compound **1** group (0.1, 0.3, 1, 3, 10, 30 μM); (5) Hypoxia/reoxygenation + positive drug group (Nimodipine, 10 μM).

Hypoxia/reoxygenation (H/R) treatment: The astrocytes were maintained at 37 °C in a humidified incubator containing 95% air and 5% CO_2_ (referred to as normoxic conditions). Hypoxic conditions were attained by exposure to 95% N_2_ and 5% CO_2_ gas mixture in a humidified incubator for 6 h and reoxygenation was achieved with normoxia conditions for another 24 h.

### 3.4. CCK-8 Assay

We prepared 100 μL of mouse astrocyte suspension at a density of 1 × 10^4^ cells/mL in a 96-well plate. The plates were preincubated for 6 h in an incubator (37 °C, 5% CO_2_). After treatment, the plates were incubated in the incubator for 24 h. Then, 10 μL of CCK8 solution was added to each well and incubated for another 1 h at 37 °C. The absorbance at 450 nm was measured with a microplate reader.

### 3.5. LDH Release

We inoculated an appropriate number of cells into a 96-well cell culture plate according to the size and growth rate of the cells so that the cell density did not exceed 80–90% when tested. The culture medium was aspirated and washed once with PBS. The fresh culture medium was replaced, and each culture well was grouped according to the experiment. Appropriate drug treatment was given according to the experimental needs, and the routine culture was continued. One hour before the scheduled detection time point, the cell culture plate was taken out from the cell incubator and incubated with the cell lysate provided by the kit to the “sample maximum enzyme activity control well”; the additional amount was 10% of the original culture medium volume. After adding the cell lysate, the solution was mixed by pipetting several times and then incubated in the cell incubator. After reaching the predetermined time, the cell culture plate was centrifuged at 400× *g* for 5 min in a multi-well plate centrifuge. A 120 μL of the supernatant was taken from each well and placed in a corresponding well of a new 96-well plate, after which sample measurements were performed.

### 3.6. Medium Glucose Content

A 5 μL volume of standard or sample was placed into a PCR tube (e.g., FTUB322/FTUB323 BeyoGold™ PCR tube (0.2 mL, raised cap, clear)) or PCR 8-strip tube (e.g., FTUB328/FTUB329 BeyoGold™ PCR 8-stripe tube (e.g., 0.2 mL, raised lid, transparent)) or 96-well PCR plate. NOTE: if the glucose concentration in the sample was very low, 20 μL of standard or sample was taken; if the sample volume was very small and the glucose concentration was in the proper range, 1–2 μL or even smaller volume of standard or sample was taken. A 185 μL Glucose Assay Reagent was used to make a final volume of 190 µL. NOTE: if 20 μL of standard or sample was added in the previous step, 170 μL of o-toluidine detection reagent was added to make the final volume 190 μL; if 1 μL of sample or standard was added in the previous step, 189 Μl of o-toluidine detection reagent was added. The final volume 190 μL was made; if other volumes of samples or standards were added in the previous step, an o-toluidine detection reagent was added to make the final volume 190 µL. After vortexing, we centrifuged at 5000× *g* for several seconds to allow the liquid to settle to the bottom of the tube. We heated at 95 °C for 8 min on a PCR machine and then cooled down to 4 °C. After cooling to 4 °C, the PCR tube was removed. We aspirated 180 μL per tube into a clean 96-well plate. Absorbance was measured at 630 nm. The glucose concentration in the sample was calculated from the standard curve.

### 3.7. ROS Release

To prepare a stock solution, 13 µL DMSO was added to 50 µg MitoSOX Red Mitochondrial Superoxide Indicator and mixed to make a 5 mM stock solution. To prepare a 5 µM working solution, the above MitoSOX Red Mitochondrial Superoxide Indicator 5 mM stock solution was diluted 1000-fold with a suitable buffer (e.g., HBSS containing Ca^2+^, Mg^2+^) to a final concentration of 5 µM. A 1–2 mL probe working solution was added to fully cover the growing cells and incubated at 37 °C for 10 min in the dark. The cells were then washed 3 times with appropriate prewarmed buffer. An appropriate counterstain solution was selected to counterstain the cells and observed after mounting.

### 3.8. ATP Content

The culture medium was removed by suction, and the lysis solution was added according to the ratio of adding 200 microliters of lysis solution to each well of the 6-well plate (equivalent to 1/10 of the volume of 2 mL of cell culture solution) to lyse the cells. When lysing cells, in order to fully lyse, a pipette was used to repeatedly pipet or shake the culture plate to make the lysate fully contact and lyse the cells. After lysis, centrifugation at 12,000× *g* at 4 °C was performed for 5 min, and the supernatant was taken for subsequent assays. A 100 microliter volume of ATP detection working solution was added to the detection well or detection tube and left at room temperature for 3–5 min. After adding 20 microliters of sample or standard to the test well or test tube, we quickly mixed it with a gun (micropipette) and measured the RLU value with a luminometer after an interval of at least 2 s.

### 3.9. Mitochondrial Oxidative Respiratory Chain Function

According to the grouping, 100 μL of 10^5^/mL primary mouse brain astrocytes suspension was mixed with 150 mL of growth medium and incubated at 37 °C. A 175 μL volume of the original growth medium was then aspirated and rinsed two times with 600 μL of hippocampal special test medium. Finally, 450 ul to 525 μL was added to observe the continuity of cells in each well under a microscope; the solution was then placed in a non-CO_2_ incubator for 1 h. Next, 75 μL of drugs were added to the four dosing tanks in each well according to the experimental design. The storage concentration and use concentration of various respiratory chain inhibitors: oligomycin storage concentration was 2.5 mmol/L, and used concentration was 1 μM/L; FCCP storage concentration 2.5 mmol/L, using concentration 1 μM/L; Antimycin A (Antimycin A) storage concentration 2.5 mmol/L, using concentration 1 μM/L. OCR, ECAR, and OCR/ECAR values were automatically calculated and recorded by the Seahorse XF 24 software

### 3.10. Apoptosis Analysis

The cell culture plate was washed twice with PBS, trypsinized, and centrifuged at 1000 rpm for 5 min, after which the culture medium was discarded. Then, cells were washed with 10 mL of PBS twice. A total of 1 × 10^5^ cells into 100 μL of 1 × Binding Buffer were then placed into a 1.5 mL EP tube and incubated with 5 μL of Annexin V for 20 min and 3 μL of PI for 10 min at room temperature in dark. A 400 μL Binding Buffer was then added, and cell apoptosis was analyzed by flow cytometry.

### 3.11. Western Blotting

Western blotting was performed using standard methods. After homogenization and centrifugation, the total protein value of the supernatant was collected. The amount of protein was determined with the BCA protein assay kit. Fifty micrograms of protein from each sample were loaded onto sodium dodecyl sulfoxide polyacrylamide gel electrophoresis (SDS-PAGE). The membrane was then transferred to a plate containing TBST solution, destained at room temperature, and blocked by shaking slowly on a shaker for 2 h. Antibody reactions were performed after blocking nonspecific binding sites with 5% bovine serum albumin. Blocked membranes were incubated with the primary antibody overnight at 4 °C, after which the membrane was washed with TBST and incubated with a secondary antibody conjugated with horseradish peroxidase. After 3 washes, proteins were visualized by enhanced chemiluminescence detection. Blots were detected using a Tanon 4600 instrument.

### 3.12. Materials

Electric homogenizer, Polytron PT 3100 (Bohemia, NY, USA); UV-vis spectrophotometer, Beckman, (Indianapolis, Indiana, USA); Electrophoresis apparatus, Bio-Rad (Hercules, CA, USA); Gel imaging analysis system (Gel Doc 2000), Bio-Rad (Hercules, CA, USA); Microplate Reader, Molecular Devices (San Jose, California, USA); NaCl, Zhiyuan Chemical Reagent Co., Ltd. (Tianjin, China); KCl, Hengxing Chemical Reagent Manufacturing Co., Ltd. (Tianjin, China); NaF and NaH_2_PO_4_, Shengao Chemical Reagent Co., Ltd. (Tianjin, China); EDTA Antigen Retrieval Solution, Guge Biotechnology Co., Ltd. (Wuhan, China); Cell Counting Kit-8, Tongren Chemical, (Kumamoto Prefecture, Upper Kyushu Island, Japan); Annexin V-FITC/PI Apoptosis Detection Kit, BD Biosciences (Franklin Lake, New Jersey, USA); DCF ROS Detection Kit, Thermo Fisher (Waltham, MA, USA); ATP Assay Kit, Beyotime (Shanghai, China); Seahorse XF Test Kit, Agilent Technologies (Palo Alto, CA, USA).

### 3.13. Statistical Analysis

Data are expressed as the mean ± SD. The measured variables between the experimental and control groups were assessed using Student’s *t*-test with nonparametric data. *p* < 0.05 was considered to be statistically significant.

## 4. Conclusions

Previous studies have demonstrated that compound **1** had excellent inhibitory activity against PYGB, highlighting its potential therapeutic effect on myocardial ischemia. In this study, we established a mouse astrocyte H/R model and found that compound **1** could enhance cell viability and inhibit LDH leakage, ROS, and intracellular glucose content. At the same time, we further confirmed the protective effect of compound **1** on the hypoxia/reoxygenation injury of mouse brain astrocytes by analyzing the cellular energy metabolism, extracellular acidification rate, cellular oxygen consumption rate, and the expression of apoptosis and apoptosis-related proteins. Taken together, our results indicate that compound **1** has a potential therapeutic effect on ischemic brain injury, whose mechanism of action is at least partly due to its regulatory effect on glucose metabolism, which also provides a basis for further development of compound **1**.

## Figures and Tables

**Figure 1 molecules-27-06333-f001:**
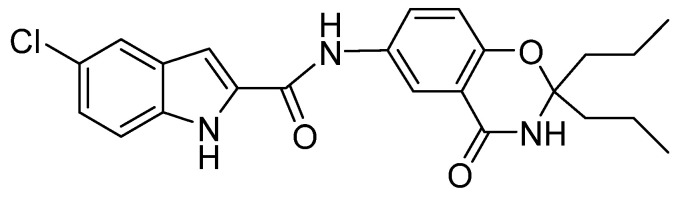
The structure of compound **1**.

**Figure 2 molecules-27-06333-f002:**
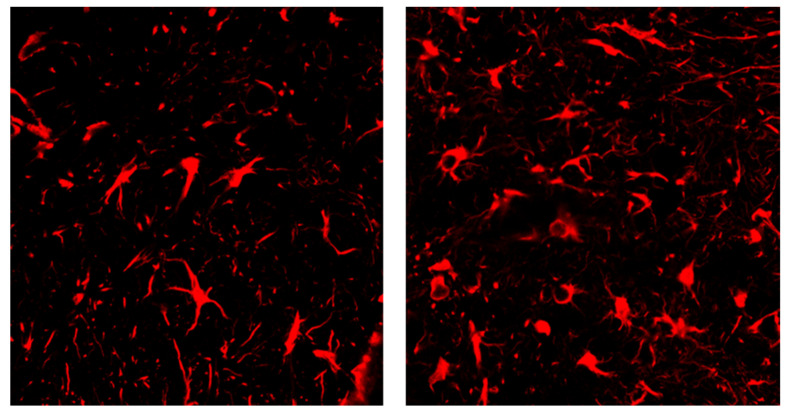
Positive expression of the immunofluorescence detection marker (GFAP).

**Figure 3 molecules-27-06333-f003:**
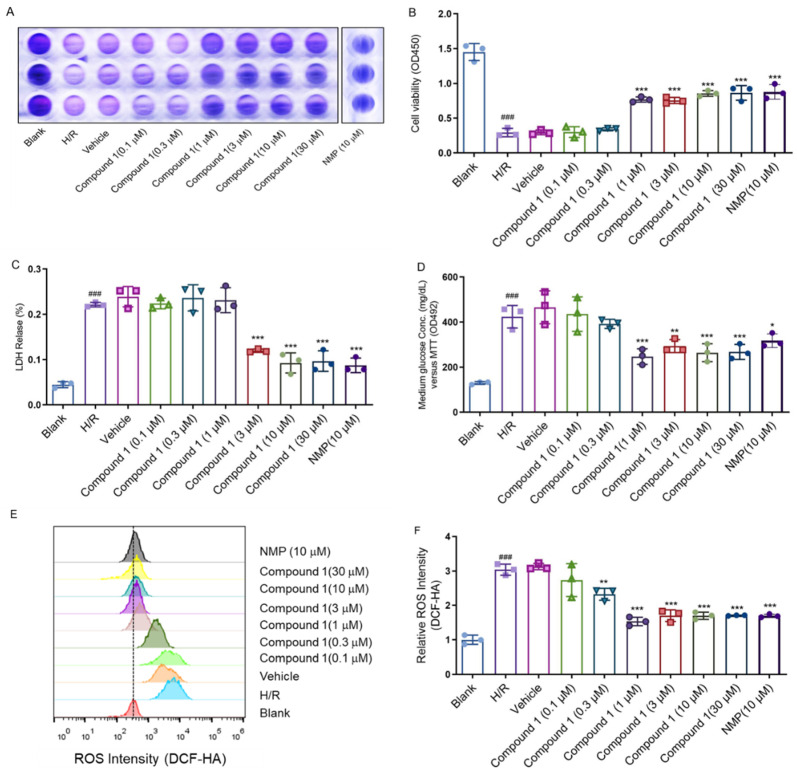
(**A,B**): Cell viability (**C**): LDH Release. (**D**): Medium glucose content: (**E**,**F**): Relative ROS intensity. Data represent mean ± SD (* *p* < 0.05, ** *p* < 0.01, *** *p* < 0.001, vs. H/R Group; ^###^
*p* < 0.001, vs. Blank Group).

**Figure 4 molecules-27-06333-f004:**
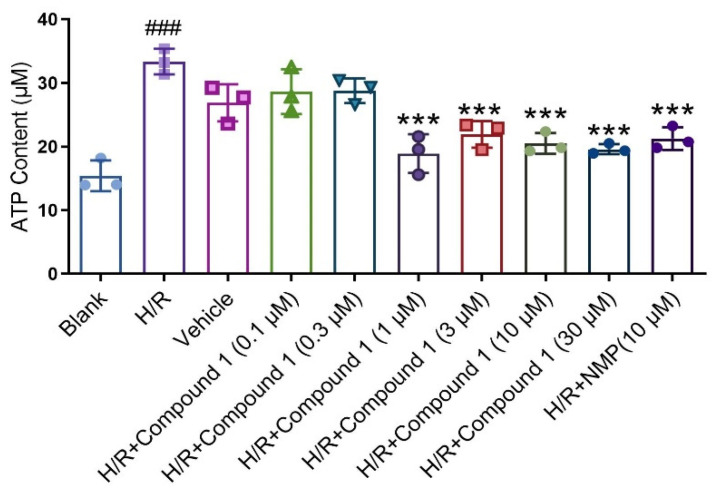
ATP Content. Data represent mean ± SD (*** *p* < 0.001, vs. H/R Group; ^###^
*p* < 0.001, vs. Blank Group).

**Figure 5 molecules-27-06333-f005:**
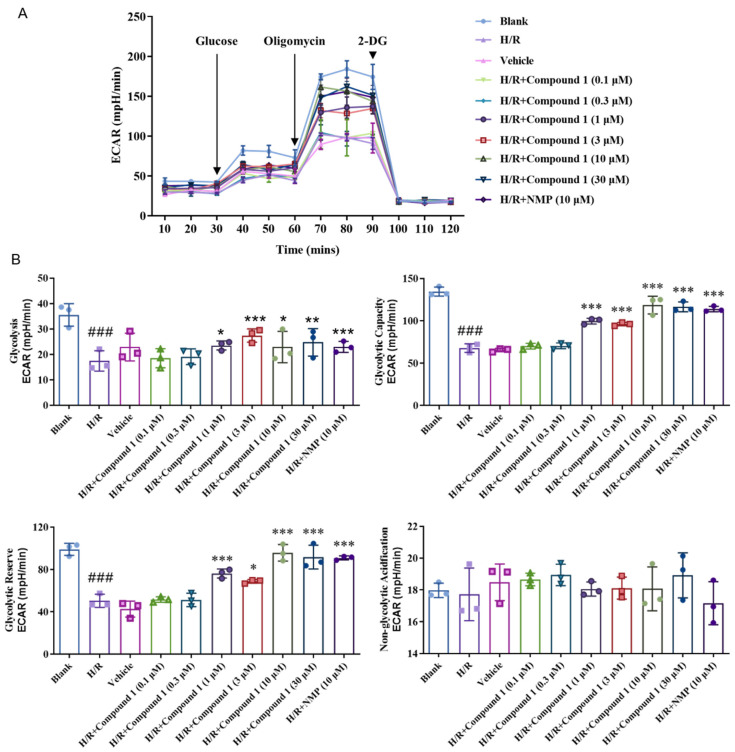
(**A**): Extracellular acidification rate (ECAR). (**B**): Bar graphs of Glycolysis, Glycolytic Capacity, Glycolytic Reserve, Non-glycolytic acidification. Data represent mean ± SD (* *p* < 0.05, ** *p* < 0.01, *** *p* < 0.001, vs. H/R Group; ^###^
*p* < 0.001, vs. Blank Group).

**Figure 6 molecules-27-06333-f006:**
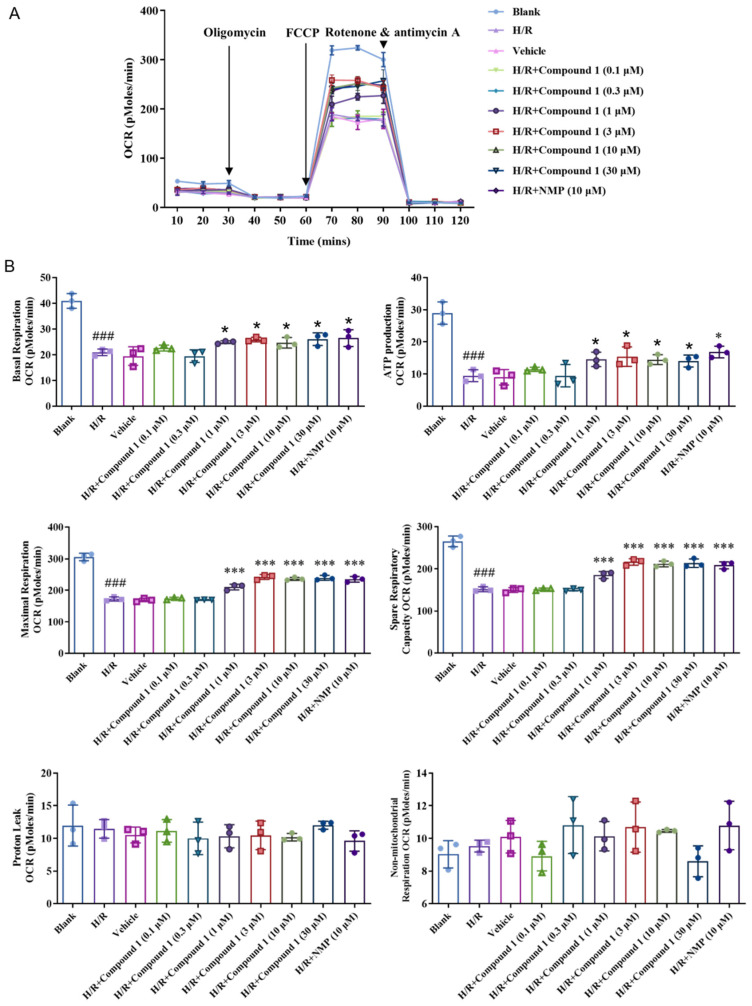
(**A**): Average oxygen consumption rate (OCR) values. (**B**): Bar graphs of Basal Respiration, ATP Production, Maximal Respiration, Spare Respiratory Capacity, Proton Leak, and Non-mitochondrial Respiration. Data represent mean ± SD (* *p* < 0.05, *** *p* < 0.001, vs. H/R Group; ^###^
*p* < 0.001, vs. Blank Group).

**Figure 7 molecules-27-06333-f007:**
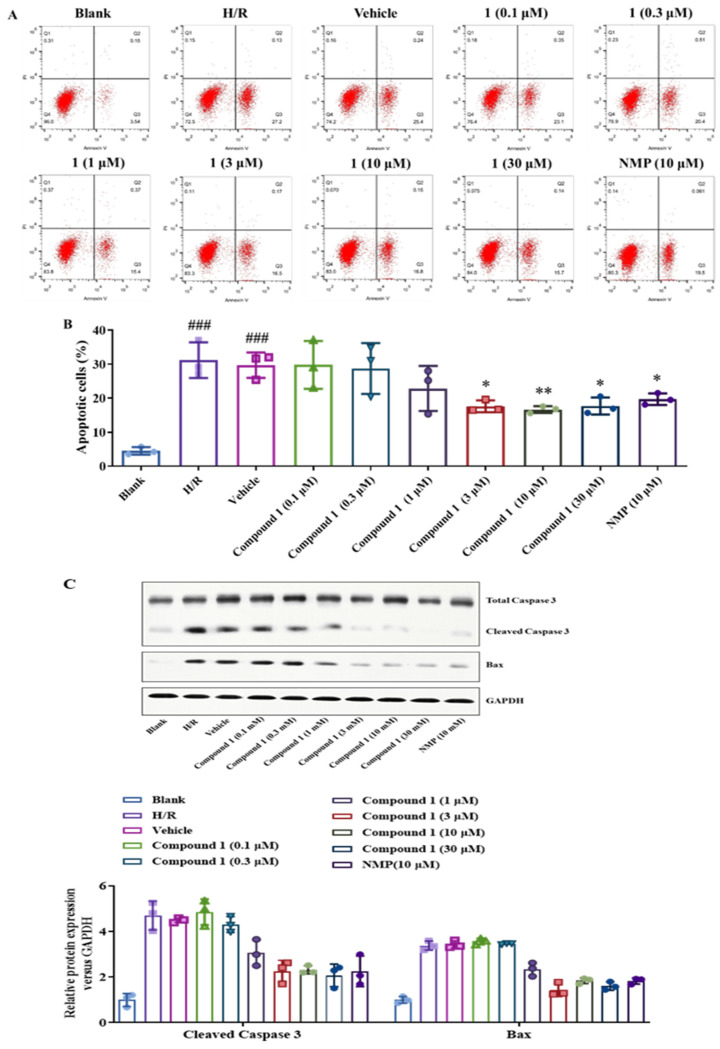
(**A**): Distribution of apoptosis measured by flow cytometry. (**B**): Bar graph of Apoptosis. (**C**): Western blot image of Total Caspase 3, Cleaved Caspase 3, Bax, GAPDH protein. Bax and Cleaved Caspase 3 protein expression. Protein content was normalized to GAPDH. Data represent mean ± SD (* *p* < 0.05, ** *p* < 0.01, vs. H/R Group; ^###^*p* < 0.001, vs. Blank Group).

## Data Availability

The data presented in this study are available on request from thecorresponding author.

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
