# Peer review of "A Novel 5-Chloro-N-phenyl-1H-indole-2-carboxamide Derivative as Brain-Type Glycogen Phosphorylase Inhibitor: Potential Therapeutic Effect on Cerebral Ischemia"

_molecules, 2022, doi:10.3390/molecules27196333_

Round 1
Reviewer 1 Report
Submitted manuscript deals that compound 1, 5-chloro-N-phenyl-1H-indole-2-carboxamide derivative ameliorates hypoxia/reoxygenation-mediated barin injury through improving antioxidative and anti-apoptotic actions. The topic of the manuscript seems to be considered as very interesting and may contribute to expand the new scientific information. However, submitted results somewhat incomplete not to execute the identification of applied cells in this study by immunostaining method with glial fibrillary acidic protein (GFAP) antibody. Accordingly, authors should explain as follows:
1. Authors should suggest additional data which relating with demonstration of astrocyte cells including morphological photographs and GFAP staining in this manuscript.
2. In this study, authors have used nimodipine, as a positive control. Please describe the special reason on the application of L-type calcium channel blocker in this study.
3. Authors applied hypoxia/reoxygenation (HR) method for the study of ischemic brain injury in this study. In general, ischemic injury with HR exhibits higher neuronal vulnerability than astrocyte cells. It takes a long time to induce astrocyte toxicity by HR only compared to both exposure of hypoxia and glucose deprivation/reoxygenation. Please describe the exact method on HR-induced astrocyte injury and special reason used HR injury in this study.
4. The number of references seems not sufficient to assert author’s opinion in this study.
5. Authors should explain the use of H9C2 cardiomyoblast cells instead of astrocyte cells in assay of mitochondrial oxidative respiratory chain function.
Reviewer 2 Report
Review Report
The manuscript intitled ‘A novel 5-chloro-N-phenyl-1H-indole-2-carboxamide derivative as brain-type glycogen phosphorylase inhibitor: Potential therapeutic effect on cerebral ischemia’ is interesting and have been written well but still need some improvement. The comments for the improvement of the manuscript are:
1. As molecule is already reported and the molecules is looks like drug candidate therefore besides the cardioprotective study it will be better that authors should also evaluate the LD50 of the molecule.
2. Figure 2, Figure 3, Figure 4 and Figure 5 need to be improved as these figures are not clear.
3. Remove the results and discussion from all the figures caption part.
The manuscript can be accepted after incorporation of the above comments.
Thanks
Dr. Syed Nazreen
Round 2
Reviewer 1 Report
Submitted manuscript was revised well according to reviewer's comments. Accordingly, this article may be acceptable in this journal.